# Association between Routine Laboratory Parameters and the Severity and Progression of Systemic Sclerosis

**DOI:** 10.3390/jcm11175087

**Published:** 2022-08-30

**Authors:** Liticia Chikhoune, Thierry Brousseau, Sandrine Morell-Dubois, Meryem Maud Farhat, Helene Maillard, Emmanuel Ledoult, Marc Lambert, Cecile Yelnik, Sebastien Sanges, Vincent Sobanski, Eric Hachulla, David Launay

**Affiliations:** 1CHU Lille, Service de Médecine Interne et Immunologie Clinique, Centre de Référence des Maladies Auto-Immunes Systémiques Rares du Nord et Nord-Ouest de France (CeRAINO), F-59000 Lille, France; 2CHU Lille, Service de Biochimie Automatisée Protéines, F-59000 Lille, France; 3U1286—INFINITE—Institute for Translational Research in Inflammation, Université de Lille, F-59000 Lille, France; 4Inserm, F-59000 Lille, France

**Keywords:** systemic sclerosis, biomarkers, interstitial lung disease, prognosis

## Abstract

(1) Background: Systemic sclerosis (SSc) is a heterogeneous connective tissue disease with a high mortality and morbidity rate. Identification of biomarkers that can predict the evolution of SSc is a key factor in the management of patients. The aim of this study was to assess the association of routine laboratory parameters, widely used in practice and easily available, with the severity and progression of SSc. (2) Methods: In this retrospective monocentric cohort study, 372 SSc patients were included. We gathered clinical and laboratory data including routine laboratory parameters: C-reactive-protein (CRP), erythrocyte sedimentation rate (ESR), complete blood count, serum sodium and potassium levels, creatinin, urea, ferritin, albumin, uric acid, N-terminal pro-brain natriuretic peptide (NTproBNP), serum protein electrophoresis, and liver enzymes. Associations between these routine laboratory parameters and clinical presentation and outcome were assessed. (3) Results: Median (interquartile range) age was 59.0 (50.0; 68.0) years. White blood cell, monocyte, and neutrophil absolute counts were significantly higher in patients with diffuse cutaneous SSc and with interstitial lung disease (ILD) (*p* < 0.001). CRP was significantly higher in patients with ILD (*p* < 0.001). Hemoglobin and ferritin were significantly lower in patients with pulmonary hypertension (PH) including pulmonary arterial hypertension and ILD associated PH (*p* = 0.016 and 0.046, respectively). Uric acid and NT pro BNP were significantly higher in patients with PH (<0.001). Monocyte count was associated with ILD progression over time. (4) Conclusions: Overall, our study highlights the association of routine laboratory parameters used in current practice with the severity and progression of SSc.

## 1. Introduction

Systemic sclerosis (SSc) is a rare autoimmune disease with a high mortality rate despite recent advances in treatment. SSc is a heterogeneous disease characterized by various forms of skin and organ involvement, mainly lung and heart [1,2]. Interstitial lung disease (ILD) and pulmonary hypertension (PH) are currently the leading causes of SSc-related mortality [3]. There are two clinical subsets according to the extent of skin fibrosis: diffuse cutaneous systemic sclerosis (dcSSc) and limited cutaneous systemic sclerosis (lcSSc). Predicting SSc outcome in terms of skin and organ progression is very important and a major challenge. Indeed, patients with progressive disease are considered to be the best candidates for a treatment aimed at stabilizing the disease [4,5]. However, well-proven prognostic factors are scarce in SSc [6]. As emphasized in several major reviews, there is still an important, unmet need for novel, validated, non-invasive biomarkers for the improvement and development of cogent and effective clinical management for patients with SSc [7,8]. Recent biomarkers found to be of interest include interleukin 6 [9]. Concerning ILD in SSc, SP-D, KL-6, and CCL18 are the most promising biomarkers, although studies on large cohorts of SSc patients to assess their role in predicting the outcome are still lacking [10,11]. In addition, increased levels of CXCL4, ICAM-1, and osteopontin have also been associated with progression of ILD in SSc patients [12,13,14]. Type I interferon (IFN) induced chemokines and the biological activity of IFN also appear promising as a means of identifying SSc patients who will progress, but data are limited [15]. The major issues are that all these biomarkers have yet to be validated and are not available in clinical practice. Conversely, some blood tests are performed on a routine basis in SSc patients’ daily care. These include C-reactive protein (CRP), erythrocyte sedimentation rate (ESR), complete blood count, serum sodium and potassium levels, creatinin, urea, ferritin, albumin, uric acid, N-terminal pro-brain natriuretic peptide (NTproBNP), serum protein electrophoresis, and liver enzymes. Several studies have suggested that these routine laboratory parameters could be of interest in assessing severity and prognosis in SSc [16,17,18,19,20]. For example, several studies have highlighted the role of CRP as a biomarker associated with the severity of SSc [16,17]. In the study by Wareing et al. [21], it was observed that a higher neutrophil count predicted a worse ILD course and higher long-term mortality in patients with SSc-ILD. However, data are scarce and these routine laboratory parameters have not been thoroughly assessed in predicting the severity and progression of SSc. Thus, the objective of our study was to fill this gap and to assess the role of simple and easily accessible laboratory parameters in predicting the severity and outcome of patients with SSc.

## 2. Materials and Methods

### 2.1. Patient Selection

This retrospective study was conducted between 2014 and 2020 in the French National Reference Center for SSc in Lille, France. Inclusion criteria were 1. Patients with SSc according to the 2013 ACR/EULAR classification criteria for SSc. 2. Age ≥ 18 years. 3. Patients with at least one visit with clinical and biological evaluation. Data protection complied with the requirements of the French National Information Science and Liberties Commission (CNIL ref. DEC18-355). For this study, patients gave oral informed consent but formal written consent was not required, in accordance with French legislation. 

### 2.2. Collected Data and Methods 

Data collected at the inclusion visit were patient demographics, history of Raynaud phenomenon (RP) and first non-RP symptom, SSc subtype, overlap with another autoimmune disease, modified Rodnan skin score (mRSS), auto antibody profile, existence of digital ulcers, organ involvement, and treatments in progress, at the time of the visit. 

Disease onset was defined as the time of onset of the first non-RP symptom. ILD was diagnosed using high resolution computed tomography (HRCT) of the lung. Pulmonary function tests including forced vital capacity (FVC) and carbon monoxide lung diffusion capacity (DLCO) were collected. Six-minute walking distance (6 MWD) was collected. PH was suspected on a Doppler echocardiogram when maximum tricuspid regurgitant jet velocity was >2.8 m/s. Pre- capillary PH was confirmed by right heart catheterization (RHC) when mean pulmonary arterial pressure (PAP) was found to be ≥25 mmHg at rest, with mean pulmonary arterial wedge pressure ≤ 15 mmHg, using the definition of PH corresponding to the inclusion period (2014–2020). Gastrointestinal tract involvement included esophageal (reflux, dysphagia, stenosis), gastric (gastric vascular ectasia), and intestinal damage (transit disorders, microbial overgrowth of the small intestine, intestinal pseudo-obstruction) and abnormal manometry and/or endoscopy test. 

### 2.3. Routine Laboratory Parameters

As we were interested in the prognostic role of routine laboratory parameters that can be used in current practice and which are easily accessible, we collected at the first inclusion visit: complete blood count (white blood cells, monocytes, neutrophils, lymphocytes, eosinophils, basophils, hemoglobin, platelets), serum sodium and potassium levels, liver enzymes (aspartate aminotransferase (AST), alanine aminotransferase (ALT), gammaglutamyltranspeptidase (GGT), alkaline phosphatase (ALP), total bilirubin), renal function (urea, creatinin), serum protein electrophoresis (alpha 1, 2, beta and gamma globulins), albumin, CRP, ESR, uric acid, ferritin, and NT pro BNP. 

First, we compared the routine laboratory parameters according to the clinical characteristics of the patients (skin extension, presence and severity of ILD, PH, and gastrointestinal tract involvement). Then, we studied whether the biological assessment at baseline could predict the evolution of SSc after a median follow-up of 12 (9.0; 15.0) months. Disease progression was defined as follows: 

▪At the skin level, a patient was considered to have disease progression if there was an increase in the Mrss > 5 points and ≥25% compared to the baseline measurement [8]. We calculated the relative change of mRSS as follows: ([mRSS at T1-mRSS at T0]/[mRSS at T0]) × 100.▪At the pulmonary level, a patient was considered to have disease progression if he or she experienced a relative decrease in FVC ≥ 10%, or a relative decline in FVC of 5–9% in association with a relative decline in DLCO of ≥15% [22]. We calculated the relative change of FVC and DLCO as follows: ([FVC at T1-FVC at T0]/[FVC at T0]) × 100 and ([DLCO at T1-DLCO at T0]/[DLCO at T0]) × 100.

### 2.4. Statistical Methods

Continuous variables are reported as median (interquartile range, IQR). Categorical variables are reported as frequency (percentage). Normality of distributions was assessed using histograms and using the Shapiro-Wilk test. Comparisons of laboratory parameters at baseline and clinical characteristics of the patients such as SSc subset, PH, gastrointestinal tract involvement, and skin and pulmonary progression were evaluated using the Mann Whitney U test. Comparisons of each laboratory parameter and ILD after adjustment for type of SSc were assessed using logistic regression models. Correlations of each routine laboratory parameter with clinical presentation and outcome of SSc (mRSS, FVC, and DLCO at baseline and their evolution between T1 (12 (9.0; 15.0) months) and baseline) were evaluated using Spearman’s correlation coefficient. For each clinical presentation and outcome of SSc, independent laboratory parameters were investigated using multivariable linear regression analyses considering as candidate variables the laboratory parameters associated in bivariate analyses at *p* < 0.20. Absence of co-linearity between candidates’ laboratory parameters was verified by calculating the variance inflation factors (VIFs). In case of co-linearity between laboratory parameters, the most clinically relevant laboratory parameters were retained as candidate variables. Full models were simplified by using a backward stepwise selection procedure (with *p* < 0.05 as selection criterion) and considering age, sex, disease duration, and type of SSc as forced variables (pre-specified confounding factors). All statistical tests were done at the two-tailed α-level of 0.05 using the SAS software version 9.4 (SAS Institute, Cary, NC, USA). 

## 3. Results

### 3.1. Characteristics at Baseline and at Follow Up

Three hundred seventy-two patients (308 females and median age of 59.0 (50.0; 68.0) years) were included. Seventy-three patients had a dcSSc (19.6%). One hundred forty-six (39.4%) patients had ILD. Twelve patients (3.2%) had PH, including eight with pulmonary arterial hypertension (PAH) and four with ILD-associated PH. Gastrointestinal tract involvement was present in 311 patients (83.6%). Baseline characteristics are presented in Table 1 and routine laboratory parameters are presented in Appendix A. Follow up data at T1 (12.0 (9.0; 15.0)) months are shown in Appendix A.

### 3.2. Routine Laboratory Parameters as Biomarkers of Severity in Systemic Sclerosis 

#### 3.2.1. Skin Involvement 

We found that the white blood cell, monocyte, and neutrophil counts were significantly higher in patients with dcSSc than in patients with lcSSc (7.3 (6.1; 8.9) vs. 6.4 (5.3; 7.8) G/L, *p* = 0.001; 0.6 (0.5; 0.8) vs. 0.6 (0.4; 0.7) /L, *p* < 0.001; 4.8 (3.5; 5.8) vs. 3.8 (3.0; 4.8) G/L, *p* < 0.001, respectively). A significant but weak positive correlation was also found between the level of monocytes and mRSS in univariate analysis. This result was not significant in multivariate analysis.

We observed a significantly lower eosinophil count in patients with dcSSc (0.1 (0.1; 0.2) vs. 0.2 (0.1; 0.3) G/L, *p* = 0.010) and a higher alpha1 globulin level (3.0 (2.7; 3.4) vs. 2.9 (2.6; 3.2) g/L, *p* = 0.023). CRP level (3.0 (3.0; 6.0) vs. 3.0 (3.0; 4.0) mg/L, *p* = 0.064), but not ESR, tended to be higher in patients with dcSSc. These results are shown in Table 2. We found a significant but weak negative correlation between albumin level and mRSS in both univariate and multivariate analysis. All correlations between laboratory parameters and mRSS at baseline are shown in Appendix A.

#### 3.2.2. Lung Involvement

We found that the white blood cell, monocyte, and neutrophil counts were significantly higher in patients with ILD than in patients without (7.1 (5.8; 8.8) vs. 6.3 (5.1; 7.6), *p* < 0.001; 0.6 (0.5; 0.8) vs. 0.5 (0.4; 0.6) /L, *p* < 0.001; 4.3 (3.5; 5.6) vs. 3.7 (2.9; 4.6) G/L, *p* < 0.001, respectively). These results are shown in Table 2 and were adjusted for the type of SSc.

We observed a significant but weak negative correlation of white blood cell, monocyte, and neutrophil counts with FVC. These results were not significant in multivariate analysis.

CRP level was significantly higher in the ILD group (3.0 (3.0; 6.0) vs. 3.0 (3.0; 4.0) mg/L, *p* < 0.001) and negatively correlated with FVC in both univariate and multivariate analysis. We observed a significant but weak positive correlation between albumin level and FVC. This result was not confirmed in multivariate analysis. A significant but weak negative correlation was found between each of the following and FVC beta globulins (in both univariate and multivariate analysis), gamma globulins (in both univariate and multivariate analysis), GGT (only in univariate analysis), alpha 1 and 2 globulins (only in univariate analysis), and ESR (only in univariate analysis). These results are shown in Appendix A.

We observed a significant but weak negative correlation between each of the following and DLCO: white blood cell (in univariate analysis only), monocyte (in both univariate and multivariate analysis), and neutrophil counts (in both univariate and multivariate analysis), CRP (univariate analysis only), uric acid (not confirmed by multivariate analysis), NTproBNP (in both univariate and multivariate analysis), ALP, GGT, alpha 1 and alpha 2 globulins (not confirmed by multivariate analysis). We also observed a significant but weak positive correlation between albumin level and DLCO. This result was confirmed by multivariate analysis.

Hemoglobin (both in univariate and multivariate analysis) and total bilirubin levels (not confirmed by multivariate analysis) were positively correlated with DLCO. These results are shown In Appendix A.

#### 3.2.3. Patients with Pulmonary Hypertension versus Patients Without

We found that uric acid and NT pro BNP level were significantly higher in patients with PH than in patients without (59.0 (52.5; 72.5) vs. 48.0 (38.0; 56.0) mg/L, *p* = 0.012; 608.5 (291.0; 1449) vs. 89.5 (53.0; 185.5) ng/L, *p* < 0.001, respectively). Hemoglobin and ferritin levels were also significantly lower in patients with PH (12.1 (10.6; 13.2) vs. 13.3 (12.5; 14.1) g/dL, *p* = 0.016; 32.5 (14.0; 56.0) vs. 73.5 (34.0; 142.0) ng/mL, *p* = 0.046, respectively). These results are shown in Table 2.

#### 3.2.4. Patients With Gastrointestinal Tract Involvement versus Patients Without

We found that total bilirubin and ferritin were lower in patients with gastrointestinal tract involvement (4.0 (3.0; 5.0) vs. 4.0 (3.0; 6.0) /L, *p* = 0.025; 64.0 (28.0; 132.0) vs. 101.0 (73.0; 187.0) ng/mL, *p* < 0.001, respectively). These results are shown in Table 2.

### 3.3. Routine Laboratory Parameters as Predictive Biomarkers of Progression in Systemic Sclerosis 

#### 3.3.1. Skin Progression

Only six patients fulfilled the criteria for skin progression and thus we were not able to assess the predictive value of the routine laboratory parameters.

We observed a significant but weak positive correlation of monocyte count and NT pro BNP level with the relative change of mRSS. These results were not confirmed by multivariate analysis and are shown in Appendix A.

#### 3.3.2. Lung Progression

According to the definition of disease progression in the Patients and Methods section, 19 SSc patients with ILD at baseline were classified as progressors and 121 as non progressors (six patients had a missing FVC value (one at baseline and five at follow up) precluding their classification as either progressors or non progressors). We found that monocytes were significantly higher in patients with progressive ILD (0.7 (0.6; 1.0) vs. 0.6 (0.5; 0.8) *p* = 0.028). These results are shown in Table 3 and in Figure 1.

We observed a significant but weak positive correlation of albumin level and platelet counts with relative variation of DLCO (in both univariate and multivariate analysis). These results are shown in Appendix A.

All these results are summarized in Table 4.

## 4. Discussion

In this study, we assessed routine laboratory parameters as potential biomarkers of severity and outcome in SSc. The main results of our study are as follows: 1. Monocyte blood count was associated with skin severity as well as with the presence, severity, and progression of ILD. 2. CRP was associated with the presence and severity of ILD while ESR was associated with severity only. 3. Elevated levels of uric acid and NT pro BNP as well as decreased hemoglobin and ferritin level were associated with the presence of PH. 4. Hypoalbuminemia was associated with skin and pulmonary severity.

Our first result is that monocytes blood count was associated with the severity of skin involvement as well as with the presence, severity, and progression of ILD. Previous studies have shown a significantly higher level of circulating monocytes expressing CD16 + in SSc patients with severe skin and lung fibrosis [23]. CD16 + circulating monocytes are thought to be the precursors of M2 tissue macrophages, which secrete CCL18, PDGF, and TGF b. Trombetta et al. [24] showed, through a wide flow cytometry surface marker analysis, that higher circulating mixed M1/M2 monocyte/macrophage cell percentages are associated with ILD, elevated systolic PAP (sPAP), and anti- topoisomerase antibody positivity in SSc. Karampitsakos et al. [25] studied the prognostic performance of complete blood count parameters in idiopathic pulmonary fibrosis (IPF) and demonstrated that peripheral blood monocyte count was predictive of all-cause mortality in patients with IPF. They also showed that patients with an elevated monocyte count exhibited more advanced disease at initial assessment when compared to patients with low levels. Yayla et al. [26] showed that a higher monocyte count independently predicted higher EUSTAR and Medsger scores (standardized β = 0.563; (*p* = 0.017) and standardized β = 0.248; (*p* = 0.051), respectively). The results of our study are consistent with the literature. Taken together, these data suggest that circulating blood monocytes are interesting biomarkers of skin and lung involvement, which could be explained by a pathophysiological role of monocytes both at the circulating level and in tissue [23].

Concerning inflammatory biomarkers, we found that CRP levels were associated with the presence and severity of ILD, while ESR was associated with severity only. Previous studies have highlighted the role of CRP as a biomarker in SSc. For example, Skaug et al. [10] and Muangchan et al. [16] showed that CRP was associated with the presence of extensive skin involvement and correlated with lung severity. Liu et al. [17] and Distler et al. [22] reported that CRP was predictive of lung evolution. CRP production is dependent on IL-6 and tocilizumab was recently approved in the USA for SSc patients with ILD to prevent further deterioration. ESR was part of the European Scleroderma Trials and Research Group (EUSTAR) 2011 score as a significant correlation was found in the univariate linear regression analysis between ESR and disease activity. However, ESR is no longer included in the 2016 revised EUSTAR activity score, as no significant association was found in the multivariate linear regression analysis [27]. As a first-line routine laboratory examination in many diagnostic procedures, ESR has many limitations, however. Physiological factors or non-inflammatory situations such as hypergammaglobulinemia may increase the ESR. The combination of CRP and fibrinogen pair makes it possible at low cost to confirm the inflammatory origin of a high ESR. In our study, gamma globulin level was not associated with the presence and outcome of skin and lung involvement. We did not assess fibrinogen level. Further studies should be done to assess the role of fibrinogen as a biomarker in SSc.

Regarding PH, we found that elevated levels of uric acid and NT pro BNP were associated with its presence. A similar result has already been described in PH. In our study, we assessed the prognostic role of these biomarkers beyond PH, including skin damage and ILD-SSc. We did not find any association of these two biomarkers with the presence and outcome of skin and lung involvement. These two biomarkers are included in the DETECT score, which predicts the risk of developing PH. Previous studies have shown a significantly higher level of uric acid and NT pro BNP in SSc patients with PH [28,29,30]. Furthermore, in a prospective study, Allanore et al. [31] also identified a decreased DLCO/VA ratio and an increased NT-pro-BNP level as predictors of PH in SSc. Savale et al. [32] reported an increased uric acid level in patients with PH which was correlated with survival after PH therapy initiation and might therefore be used as a noninvasive indicator of disease severity. This could be explained by a disturbance in the production and metabolism of uric acid in remodeled pulmonary arteries in PH patients.

We found that patients with PH had a significantly lower level of hemoglobin and ferritin at baseline. This has already been reported elsewhere. For examples, Tezcan et al. [33] found a significantly lower level of hemoglobin in SSc with PH patients (12.2 ± 2 vs. 13 ± 1.8 g/dL, *p* = 0.0015). In the prospective cohort study conducted by Hsu et al. [34], it was reported that anemia was predictive of poor outcome in SSc patients at risk for PH. Anemia in SSc patients could be explained by malabsorption, iron deficiency, and gastrointestinal blood loss. A growing body of evidence suggests a central role of iron homeostasis and anemia in the pathophysiology, progression, and prognosis of PH [35,36]. The study conducted by Ruiter et al. [36] showed that the prevalence of iron deficiency was higher in SSc patients with PH than in those without (46 versus 16%, respectively) and that the survival rate was lower in SSc-PH patients with iron deficiency than in those without iron deficiency (*p* = 0.03). Yet, the precise mechanism of iron deficiency is still poorly understood. Given that many SSc patients have a wide range of gastrointestinal symptoms, it is plausible that they have a lower iron intake or absorption. In the same study, Ruiter et el. found a high level of hepcidin in patients with SSc. Hepcidin production by the liver is dependent on pro-inflammatory cytokines such as IL-6. The production of hepcidin by other cells such as monocytes was mentioned in the review by Theurl et al. [37]. It is important to detect iron deficiency and anemia (with or without iron deficiency) as they may constitute a potential therapeutic target in order to improve morbidity and mortality in patients with PH [35,38]. However, this result needs to be confirmed in larger SSc cohorts to support its relevance since only 12 patients had PH in our study. 

We also found a significant association between albumin level and skin and pulmonary severity in SSc. A few studies have investigated the role of albumin as a biomarker associated with the severity of SSc, and these were mainly interested in the prevalence of and potential association with malnutrition (whether or not taking albuminemia into account) in SSc patients. For example, in the study conducted by Wojteczek et al. [39], 16.1% of SSc patients had hypoalbuminemia < 35 g/L. In the study by Caimmi et al. [40], malnutrition was diagnosed in 9.2% of SSc patients (95% CI, 4.4–14.0%) and was associated with worse gastrointestinal symptoms (*p* = 0.007), worse predicted DLCO/VA and FVC (*p* = 0.009, respectively), and worse disease severity according to the Medsger severity score (*p* < 0.001). They found that lung involvement was significantly associated with nutritional status. A link between malnutrition and muscle weakness can be hypothesized, affecting the performance on the pulmonary function test. Hypoalbuminemia can also be explained by the presence of a biological inflammatory syndrome. Nevertheless, these results must be interpreted with caution because of the weak correlations found and they need to be supported by prospective studies.

Our study has some limitations: (1) the study was monocentric and retrospective, with the associated bias; (2) the frequency of skin progression was too low in our cohort to allow a statistical analysis of routine laboratory parameters as skin prognosis biomarkers; (3) we did not assess fibrinogen; (4) only 12 patients in our study had PH, making it difficult to interpret the results; (5) one of the limits of our study is the multiplicity of statistical tests carried out, thereby increasing the alpha risk; (6) our study did not include a validation cohort; (7) our study was not designed to build a composite score of progression to analyze at the individual level the risk of worsening for a given patient. Indeed, this study was not designed to establish a risk score and thus does not provide a ready-to-use set of prognostic biomarkers. Nevertheless, it opens interesting avenues for identifying patients at high risk of progression. For example, a patient with newly evolving SSc, and with an initial high monocyte count, should be given special consideration due to the risk of pulmonary progression. While the majority of the values of the routine laboratory parameters studied were within the norm, our study shows that even a moderate elevation, for example of CRP, could be interesting in SSc because of its association with disease characteristics and/or progression.

Our study has some strengths, including the high number of well-phenotyped patients with available laboratory data and the use of simple biomarkers which can easily be used in everyday practice. To our knowledge, this is one of the first studies to study several different routine laboratory parameters simultaneously.

## 5. Conclusions

Our results may help identify patients with SSc having severe skin and lung manifestations and outcome. This has important clinical implications as inexpensive, non-invasive tests used in routine practice can be used to identify high-risk patients who should undergo more careful evaluation and for whom early treatment would be indicated. However, this needs to be supported by prospective controlled studies. 

## Figures and Tables

**Figure 1 jcm-11-05087-f001:**
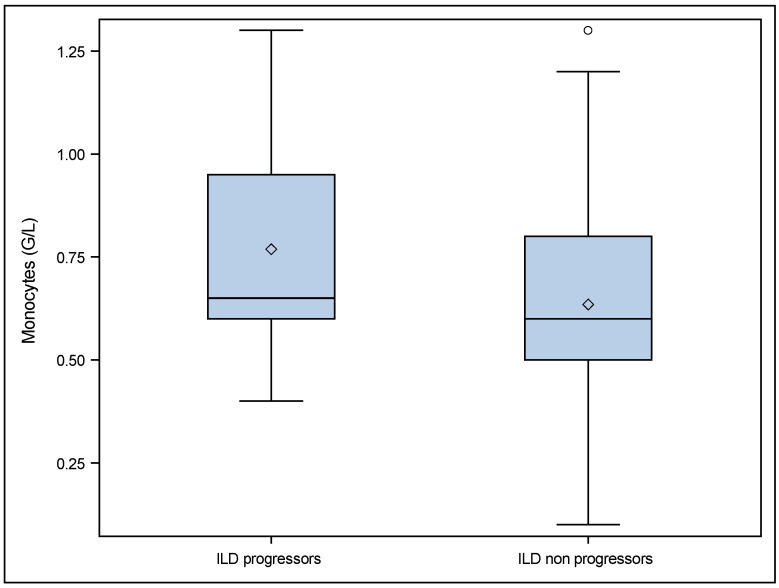
Comparisons of monocytes at baseline in ILD progressors and ILD non progressors. ° *p* value = 0.028. ILD progressors: patients with ILD at baseline who were progressors during follow up (19/146). ILD non progressors: patients with ILD at baseline who were non progressors during follow up (121/146). 6 patients had missing FVC data. The diamond corresponds to the values presented as median (IQR).

**Table 1 jcm-11-05087-t001:** Patients’ baseline characteristics.

Characteristics	N*. with Available Data n (%)	SSc Patients Presented as Frequency (Percent), or Median (Interquartile Range)
**Demographics**		
Sex, female	372 (100)	308 (82.8)
Age, years	372 (100)	59.0 (50.0; 68.0)
BMI, kg/m^2^	372 (100)	24.5 (21.9; 28.0)
**Disease characteristics**		
Cutaneous subset, diffuse	372 (100)	73 (19.6)
Disease duration ^#^	372 (100)	
≤3 years		92 (24.7)
>3 years		280 (75.3)
**Antibody status **		
Anti-nuclear antibodies	372 (100)	365 (98.1)
Anti-centromere		197 (53)
Anti-topoisomerase I		82 (22)
Anti-RNA polymerase III		12 (3.2)
Anti RNP		11 (3)
**Organ involvement**		
Interstitial lung disease	371 (99)	146 (39.4)
Limited ILD *^£^*Extensive ILD *^£^*		103 (71)
		42 (29)
FVC%	371 (99)	104.0 (91.0; 115.0)
DLCO%	363 (97)	70.0 (59.0; 81.0)
mRSS	372 (100)	2.5 (1.0; 6.0)
Gastrointestinal tract	372 (100)	311 (83.6)
Pulmonary hypertension	372 (100)	12 (3.2)
Renal crisis	364 (97)	2 (0.5)
Digital ulcers	370 (99)	168 (45.4)
Raynaud phenomenon	372 (100)	370 (99.5)
Overlap syndrome	372 (100)	129 (34.7)
**Current treatment**		
Glucocorticoids	193 (52)	64 (33.2)
Immunosuppressive drugs ^$^	193 (52)	48 (24.9)

Values are presented as frequency (percent), median (interquartile range). N* number, SSc systemic sclerosis, BMI body mass index. ^#^ Disease duration defined from the first symptom outside the Raynaud phenomenon, FVC% forced vital capacity (% predicted value), DLCO% carbon monoxide lung diffusion capacity (% predicted value). RNA Ribonucleic acid. RNP Ribonucleoprotein. ^$^ Immunosuppressive drugs used were Cyclophosphamide, mycophenolate mofetil, azathioprine, methotrexate, anti-calcineurin, intravenous immunoglobulins, rituximab, anti TNFalpha, tocilizumab (no patients received tocilizumab). ^£^ Limited ILD: Limited involvement of less than 10% of pulmonary parenchyma; Extensive ILD: reaching more than 30% of pulmonary parenchyma. In patients with HRCT extent of 10–30% (termed indeterminate disease), an FVC threshold of 70% was an adequate prognostic substitute, according to Goh et al. ATS journal, 2008. During follow-up, 32 patients (8.6%) died.

**Table 2 jcm-11-05087-t002:** Routine laboratory parameters as biomarkers of severity in systemic sclerosis.

	dcSSc (N = 73)	lcSSc (N = 299)	P	ILD (N = 146)	No ILD (N = 225)	P *	PH (N = 12)	No PH (N = 360)	P	Gastrointestinal involvement (N = 311)	No Gastrointestinal Involvement (N = 61)	P
White blood cells G/L	7.3 (6.1; 8.9)	6.4 (5.3; 7.8)	**0.001**	7.1 (5.8; 8.8)	6.3 (5.1; 7.6)	**0.001**	7.0 (5.7; 9.7)	6.6 (5.5; 7.9)	0.56	6.6 (5.5; 7.9)	6.5 (5.1; 8.1)	0.58
Monocytes G/L	0.6 (0.5; 0.8)	0.6 (0.4; 0.7)	**<0.001**	0.6 (0.5; 0.8)	0.5 (0.4; 0.6)	**<0.001**	0.6 (0.4; 0.7)	0.6 (0.4; 0.7)	0.82	0.6 (0.5; 0.7)	0.6 (0.4; 0.7)	0.29
Neutrophils G/L	4.8 (3.5; 5.8)	3.8 (3.0; 4.8)	**<0.001**	4.3 (3.5; 5.6)	3.7 (2.9; 4.6)	**<0.001**	4.5 (3.7; 6.4)	4.0 (3.1; 5.0)	0.33	4.0 (3.1; 5.0)	3.9 (2.9; 5.1)	0.50
Eosinophils G/L	0.1 (0.1; 0.2)	0.2 (0.1; 0.3)	**0.010**	0.2 (0.1; 0.2)	0.2 (0.1; 0.3)	0.58	0.1 (0.1; 0.3)	0.2 (0.1; 0.2)	0.76	0.2 (0.1; 0.3)	0.1 (0.1; 0.2)	**0.079**
Lymphocytes G/L	1.6 (1.3; 2.3)	1.7 (1.2; 2.1)	0.75	1.6 (1.2; 2.3)	1.7 (1.2; 2.1)	0.34	1.7 (0.9; 2.2)	1.7 (1.2; 2.1)	0.67	1.7 (1.2; 2.1)	1.6 (1.3; 2.1)	0.73
Basophils G/L	0.0 (0.0; 0.1)	0.0 (0.0; 0.1)	0.47	0.0 (0.0; 0.1)	0.0 (0.0; 0.1)	0.26	0.0 (0.0; 0.1)	0.0 (0.0; 0.1)	0.47	0.0 (0.0; 0.1)	0.0 (0.0; 0.1)	0.62
Hemoglobin g/dL	13.1 (12.5; 14.0)	13.3 (12.5; 14.1)	0.69	13.1 (12.4; 14.0)	13.4 (12.5; 14.1)	0.47	12.1 (10.6; 13.2)	13.3 (12.5; 14.1)	**0.016**	13.3 (12.5; 14.0)	13.3 (12.8; 14.2)	0.24
Platelets G/L	264.0 (226.0; 311.0)	255.0 (215.0; 297.0)	0.33	265.0 (228.0; 307.0)	250.5 (210.0; 293.0)	0.14	236.0 (204.0; 276.0)	256.0 (216.0; 303.0)	0.24	257.0 (222.0; 305.0)	239.0 (200.0; 292.0)	0.14
CRP mg/L	3.0 (3.0; 6.0)	3.0 (3.0; 4.0)	0.064	3.0 (3.0; 6.0)	3.0 (3.0; 4.0)	**0.003**	3.0 (3.0; 5.0)	3.0 (3.0; 4.0)	0.85	3.0 (3.0; 5.0)	3.0 (3.0; 3.0)	0.24
ESR mm/h	9.0 (2.0; 19.0)	11.0 (6.0; 21.0)	0.34	11.0 (6.0; 22.0)	10.0 (6.0; 18.0)	0.62	11.0 (5.0; 32.0)	10.5 (6.0; 19.0)	0.52	11.0 (6.0; 19.0)	10.0 (6.0; 23.0)	0.79
NT pro BNP ng/L	105.5 (45.0; 309.0)	89.0 (55.0; 177.0)	0.31	98.0 (57.0; 281.0)	88.0 (53.0; 168.0)	0.26	608.5 (291.0; 1449)	89.5 (53.0; 185.5)	**<0.001**	93.0 (57.0; 206.0)	76.0 (45.0; 161.0)	0.28
Ferritin ng/mL	76.0 (32.0; 160.0)	70.0 (33.0; 135.0)	0.51	78.0 (32.0; 143.0)	68.0 (33.0; 137.0)	0.67	32.5 (14.0; 56.0)	73.5 (34.0; 142.0)	**0.046**	64.0 (28.0; 132.0)	101.0 (73.0; 187.0)	**<0.001**
Uric acid mg/L	48.0 (35.5; 57.0)	48.0 (39.0; 57.0)	0.42	48.0 (36.0; 57.0)	48.0 (40.0; 57.0)	0.50	59.0 (52.5; 72.5)	48.0 (38.0; 56.0)	**0.012**	48.0 (38.0; 56.0)	48.0 (41.0; 61.0)	0.17
AST UI/L	20.0 (17.0; 23.0)	22.0 (19.0; 26.0)	**0.007**	21.0 (18.0; 25.0)	22.0 (18.0; 26.0)	0.52	20.0 (18.0; 22.0)	22.0 (18.0; 26.0)	0.23	21.0 (18.0; 26.0)	22.0 (18.0; 25.0)	0.60
ALT UI/L	16.0 (11.0; 20.0)	16.0 (13.0; 22.0)	0.070	16.0 (12.0; 22.0)	15.5 (13.0; 21.0)	0.97	12.5 (9.0; 15.0)	16.0 (12.0; 21.5)	**0.035**	16.0 (12.0; 21.0)	17.0 (13.5; 22.0)	0.53
ALP UI/L	60.0 (51.0; 76.0)	65.0 (53.0; 86.0)	0.11	62.0 (52.0; 80.0)	66.0 (52.0; 85.0)	0.20	73.0 (56.0; 110.0)	64.0 (52.0; 83.0)	0.16	65.0 (52.0; 85.0)	63.0 (50.0; 81.5)	0.40
GGT UI/L	20.0 (12.0; 31.0)	21.0 (14.0; 39.0)	0.25	22.0 (15.0; 40.0)	20.0 (14.0; 36.0)	0.85	18.0 (14.0; 65.0)	21.0 (14.0; 37.0)	0.74	21.0 (14.0; 38.0)	20.0 (13.0; 39.0)	0.51
Total bilirubin mg/L	4.0 (3.0; 5.0)	4.0 (3.0; 5.0)	0.58	4.0 (3.0; 5.0)	4.0 (3.0; 5.0)	0.58	3.5 (3.0; 6.0)	4.0 (3.0; 5.0)	0.97	4.0 (3.0; 5.0)	4.0 (3.0; 6.0)	**0.025**
Serum sodium mmol/L	140.0 (139.0; 142.0)	140.0 (139.0; 142.0)	0.51	140.0 (139.0; 141.0)	140.5 (139.0; 142.0)	0.11	139.5 (137.0; 141.0)	140.0 (139.0; 142.0)	0.33	140.0 (139.0; 142.0)	140.0 (138.0; 141.5)	0.17
Serum potassium mmol/L	4.0 (3.9; 4.4)	4.2 (4.0; 4.4)	**0.039**	4.1 (3.9; 4.4)	4.2 (4.0; 4.4)	0.57	4.2 (3.8; 4.4)	4.2 (4.0; 4.4)	0.96	4.2 (4.0; 4.4)	4.1 (4.0; 4.4)	0.27
Urea g/L	0.3 (0.3; 0.4)	0.3 (0.3; 0.4)	0.88	0.3 (0.3; 0.4)	0.3 (0.3; 0.4)	0.75	0.4 (0.4; 0.5)	0.3 (0.3; 0.4)	**0.003**	0.3 (0.3; 0.4)	0.3 (0.3; 0.4)	0.33
Creatinin mg/L	8.0 (7.0; 9.0)	8.0 (7.0; 9.0)	0.70	8.0 (7.0; 9.0)	7.0 (7.0; 9.0)	0.21	9.0 (7.5; 10.5)	8.0 (7.0; 9.0)	**0.039**	8.0 (7.0; 9.0)	8.0 (7.0; 9.0)	0.90
A1globulins g/L	3.0 (2.7; 3.4)	2.9 (2.6; 3.2)	**0.023**	2.9 (2.7; 3.3)	2.8 (2.5; 3.2)	0.24	3.2 (2.8; 3.5)	2.9 (2.6; 3.2)	0.11	2.9 (2.6; 3.2)	2.9 (2.6; 3.2)	1.00
A2globulins g/L	7.8 (6.8; 8.4)	7.5 (6.9; 8.2)	0.31	7.7 (6.9; 8.4)	7.5 (6.8; 8.1)	0.36	7.6 (7.4; 9.0)	7.6 (6.8; 8.2)	0.43	7.6 (6.9; 8.3)	7.4 (6.5; 8.0)	0.12
Betaglobulins g/L	7.9 (7.3; 8.5)	7.7 (7.1; 8.4)	0.48	7.7 (7.2; 8.4)	7.7 (7.1; 8.4)	0.97	7.9 (7.5; 9.1)	7.7 (7.1; 8.4)	0.52	7.7 (7.1; 8.4)	7.7 (7.2; 8.4)	0.60
Gammaglobulins g/L	10.3 (7.8; 13.5)	9.8 (8.4; 11.6)	0.33	10.1 (8.0; 12.4)	9.8 (8.5; 11.9)	0.93	10.9 (10.5; 12.8)	9.9 (8.2; 12.0)	0.059	9.9 (8.2; 12.0)	10.1 (9.0; 12.8)	0.21
Albumin g/L	40.3 (38.0; 44.0)	41.0 (39.0; 43.0)	0.27	40.3 (38.0; 43.0)	41.0 (39.1; 43.0)	0.21	39.4 (37.2; 41.0)	41.0 (38.8; 43.0)	0.064	41.0 (38.5; 43.0)	41.1 (39.7; 44.0)	0.15

Values are presented as median (interquartile range [IQR]). dcSSc: diffuse cutaneous systemic sclerosis; lcSSc: limited cutaneous systemic sclerosis; ILD: interstitial lung disease; PH: pulmonary hypertension; SSc: systemic sclerosis; CRP: C-reactive protein; ESR: erythrocyte sedimentation rate; ALP: alkaline phosphatase; AST: aspartate aminotransferase; ALT: alanine aminotransferase; GGT: gammaglutamyltranspeptidase; NTproBNP: N-terminal pro-brain natriuretic peptide; G/L: Giga/Liter; A1 globulins: alpha 1 globulins; A2 globulins: Alpha 2 globulins; No: Non ILD, non PH, and non-gastrointestinal tract involvement; P: *p* value < 0.05 (significant results appear in bold in the table). * *p* values are adjusted for type of SSc.

**Table 3 jcm-11-05087-t003:** Comparison of routine laboratory parameters between ILD progressors and non progressors.

	ILD Progressors (N = 19)	ILD Non Progressors (N = 121)	*p*
White blood cells G/L	7.4 (5.8; 9.1)	7.1 (5.8; 8.8)	0.69
Monocytes G/L	0.7 (0.6; 1.0)	0.6 (0.5; 0.8)	**0.028**
Neutrophils G/L	4.5 (3.3; 6.4)	4.4 (3.5; 5.6)	0.90
Eosinophils G/L	0.2 (0.1; 0.2)	0.2 (0.1; 0.3)	0.93
Lymphocytes G/L	1.6 (1.2; 2.0)	1.6 (1.2; 2.3)	1.00
Basophils G/L	0.0 (0.0; 0.1)	0.0 (0.0; 0.1)	0.59
Hemoglobin g/dL	13.9 (12.8; 14.7)	13.1 (12.4; 14.2)	0.12
Platelets G/L	243.0 (221.0; 291.5)	273.0 (230.0; 314.0)	0.60
CRP mg/L	4.0 (3.0; 5.0)	3.0 (3.0; 5.0)	0.29
ESR mm/h	9.0 (5.5; 22.0)	10.0 (6.0; 22.0)	0.93
NT pro BNP ng/L	113.0 (67.0; 291.0)	97.0 (55.0; 248.0)	0.51
Ferritin ng/mL	99.0 (61.5; 151.0)	70.0 (29.0; 129.0)	0.094
Uric acid mg/L	48.0 (40.0; 59.0)	46.5 (35.0; 56.0)	0.59
AST UI/L	23.0 (19.5; 28.5)	21.0 (18.0; 25.0)	0.37
ALT UI/L	18.0 (13.5; 22.5)	16.0 (12.0; 22.0)	0.59
ALP UI/L	58.0 (52.5; 94.5)	62.0 (52.0; 79.0)	0.90
GGT UI/L	29.0 (22.0; 50.0)	21.5 (15.0; 43.0)	0.13
Total bilirubin mg/L	4.5 (3.0; 6.5)	3.0 (3.0; 5.0)	0.099
Serum sodium mmol/L	140.0 (139.0; 141.0)	140.0 (139.0; 141.0)	0.95
Serum potassium mmol/L	4.2 (4.0; 4.4)	4.2 (4.0; 4.5)	0.37
Urea g/L	0.3 (0.3; 0.4)	0.3 (0.3; 0.4)	0.70
Creatinin mg/L	8.0 (7.0; 9.0)	8.0 (7.0; 9.0)	0.89
A1globulins g/L	3.0 (2.8; 3.4)	3.0 (2.6; 3.3)	0.45
A2globulins g/L	8.0 (7.4; 8.6)	7.7 (6.9; 8.4)	0.64
Beta globulins g/L	7.5 (6.8; 8.3)	7.8 (7.2; 8.4)	0.29
Gamma globulins g/L	9.0 (8.0; 10.3)	10.1 (7.9; 12.2)	0.24
Albumin g/L	40.2 (38.0; 43.1)	40.1 (38.0; 43.1)	0.54

Values are presented as median (interquartile range); CRP: C-reactive protein; ESR: erythrocyte sedimentation rate; ALP: alkaline phosphatase; AST: aspartate aminotransferase; ALT: alanine aminotransferase; GGT: gammaglutamyltranspeptidase; NTproBNP: N-terminal pro-brain natriuretic peptide; G/L: Giga/Liter. *p* value < 0.05 (significant results appear in bold in the table). A1 globulins: alpha 1 globulins; A2 globulins: alpha 2 globulins; 6 patients had a missing FVC value (1 at baseline and 5 at follow up) precluding their classification as either progressors or non progressors.

**Table 4 jcm-11-05087-t004:** Summary of the main results in univariate and multivariate analysis.

	dcSSc	mRSS	ΔmRSS	PH	ILD	FVC	DLCO	ΔFVC	ΔDLCO	ILD Progressors
White blood cells (G/L)										
Monocytes (G/L)							*			
Neutrophils (G/L)							*			
Eosinophils (G/L)										
Hemoglobin (g/dL)							*			
Platelets (G/L)									*	
CRP (mg/L)						*				
ESR (mm/h)										
NTproBNP (ng/L)							*			
Uric acid (mg/L)										
Ferritin(ng/mL)										
AST (UI/L)										
ALT (UI/L)										
ALP (UI/L)										
GGT (UI/L)										
Total bilirubin (mg/L)										
Serum potassium (mmol/L)										
Urea (g/L)										
Creatinin (mg/L)										
A1globulins (g/L)										
A2globulins (g/L)										
Betaglobulins (g/L)						*				
Gammaglobulins (g/L)						*				
Albumin (g/L)		*					*		*	
Lymphocytes (G/L)										
Basophils (G/L)										
Serum sodium (mmol/L)										

dcSSc: diffuse cutaneous systemic sclerosis; mRSS: modified Rodnan skin score; pH: pulmonary hypertension; ILD: interstitial lung disease; %FVC: forced vital capacity (% predicted value); %DLCO: carbon monoxide lung diffusion capacity; CRP: C-reactive protein; ESR: erythrocyte sedimentation rate; NTproBNP: N-terminal pro-brain natriuretic peptide; AST: aspartate aminotransferase; ALT: alanine aminotransferase; GGT: gammaglutamyltranspeptidase; ALP: alkaline phosphatase. * Significant results in multivariate analysis. Δ Relative progression mRSS, FVC and DLCO.

 Positive significant association.

 Negative significant association.
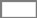
 No significant association.

 CRP was higher in patients with dcSSc but not significantly *p* = 0.064.

## Data Availability

The datasets used and/or analyzed during the current study are available from the corresponding author upon reasonable request from any qualified researcher.

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
