# Peer review of "Association between Routine Laboratory Parameters and the Severity and Progression of Systemic Sclerosis"

_jcm, 2022, doi:10.3390/jcm11175087_

Round 1

Reviewer 1 Report

Here the authors submit to multiple statistical analyses cross-sectional standard laboratory data aiming at identifying predictors of evolution in SSc.

The reader is submerged by an avalanche of data and, at least in my case, has difficulties in understanding to which extent these data have sufficient strength to be of any use in clinical practice.

Major points to be addressed.

1. Statistics. The authors mention that according to the presence or absence of normal distribution, the applied parametric or non-parametric statistics, and present the data as mean+/-SD or median and interquartile range. This variability increases the difficulty in reading the tables. I would strongly suggest to use uniformly non-parametric statistic, including when correlations are sought of.

2. Tables 1 & 2. The column most on the right requires appropriate headings in order to understand what is presented: mean, median, range, standard deviation, etc.?

3. Table 1 presents "base-line characteristics". In contrast with this definition the table includes "Death no". If I understand correctly, this refers to the number of deaths. How is it possible that at base-line there are already death individuals? Please, clarify and amend.

4. Definition of ILD extent. The authors state: "Limited involvement of less than 10% of pulmonary parenchyma and extensive ILD reaching more than 20% of pulmonary parenchyma". What about individuals with ILD involving 10 to 20% of the parenchyma? How was ILD extension quantified? Please, specify and in the case amend.

5. Table 3. When inspecting the CRP values in ILD vs nonILD one reads 3±SD vs 3±SD (p = 0.003). If I understand correctly, in the text, line 203, the reported values are (7.4 ± 17.6 vs 5.0 ± 6.8 mg/L,  p<0.001). Please, clarify and amend.

6. Informed consent. In the M&M section the authors state: For this study, patients were informed but formal written consent was not required according to French legislation. In the specific heading at the end of the manuscript the authors state: Informed Consent Statement: Informed consent was obtained from all subjects involved in the study. It seems to me that there is a contradiction between these two sentences.  I understand that "no consent was sought", since not required by French legislation. This should  be reported uniformly.

7. On line 58 one reads: "Type I interferon chemokines molecules...". Are the authors referring to IFN-induced chemokines or are they referring to the biological activity of IFN, or other. Please, clarify.

8. Importantly, in the discussion the authors should clarify what is their thinking concerning the use of the predictors they describe in the MS at the single patient level. a) On the light that the levels of the variables are most often in the normal range. b) The strengths of correlations are weak to very weak.

9. I personally think that the most useful table is the current supplementary table 6. This should go to the main text and could be improved by grading the colors according to the strength of the significance. In the case, current table 2 could go to the supplementary.

10. Line 134 "after adjustment for type of SSc". Is this adjustment or simply classification? Similarly, line 193, "p values are adjusted for type SSc." Further, line 199: "These results are shown in Table 3 and were adjusted with the type of SSc". Please, clarify.

Minors

1. The text could be improved when revised by a proficient English writer. To many are the specific terms which need intervention to allow their listing here.

2. The adverb "also" is repeated 22 times in the text. Is it really necessary?

Author Response

Major points to be addressed.

  1. Statistics. The authors mention that according to the presence or absence of normal distribution, the applied parametric or non-parametric statistics, and present the data as mean+/-SD or median and interquartile range. This variability increases the difficulty in reading the tables. I would strongly suggest to use uniformly non-parametric statistic, including when correlations are sought of.

Response 1 : We thank the reviewer for this comment. Accordingly, we have standardized the presentation of continuous variables by presenting median (IQR) for all parameters.

  1. Tables 1 & 2. The column most on the right requires appropriate headings in order to understand what is presented: mean, median, range, standard deviation, etc.?

Response 2 : We thank the reviewer for this comment. We have now clarified the headings in table 1 (page 4, line 345) and table 2 that has now become table S1 in the revised version of the manuscript (page 5 line 397). We have standardized the presentation by presenting frequency (percent) or median (IQR) for variables.

  1. Table 1 presents "base-line characteristics". In contrast with this definition the table includes "Death no". If I understand correctly, this refers to the number of deaths. How is it possible that at base-line there are already death individuals? Please, clarify and amend.

Response 3 : We thank the reviewer for pointing out this inconsistency. The reference to the number of deaths has now been deleted from the baseline characteristics table (page 4, line 345).

  1. Definition of ILD extent. The authors state: "Limited involvement of less than 10% of pulmonary parenchyma and extensive ILD reaching more than 20% of pulmonary parenchyma". What about individuals with ILD involving 10 to 20% of the parenchyma? How was ILD extension quantified? Please, specify and in the case amend.

Response 4 : We thank the reviewer for this important comment. Goh et al. (ATS journal, 2008) defined limited ILD as <10% lung parenchymal involvement and extensive involvement as >30% lung parenchymal damage. In patients with HRCT extent of 10– 30% (termed indeterminate disease), an FVC threshold of 70% was an adequate prognostic substitute, which was the definition we used for our work but was misleadingly presented in the first version of the manuscript. We have modified the manuscript accordingly (page 5, line 395).

  1. Table 3. When inspecting the CRP values in ILD vs non ILD one reads 3±SD vs 3±SD (p = 0.003). If I understand correctly, in the text, line 203, the reported values are (7.4 ± 17.6 vs 5.0 ± 6.8 mg/L, p<0.001). Please, clarify and amend.

Response 5 : We thank the reviewer for pointing this out. We have modified the table 3 currently table 2 on the revised version of the manuscript on page 6. We have standardized the presentation of continuous variables by presenting median (IQR) for all parameters.

  1. Informed consent. In the M&M section the authors state: For this study, patients were informed but formal written consent was not required according to French legislation. In the specific heading at the end of the manuscript the authors state: Informed Consent Statement: Informed consent was obtained from all subjects involved in the study. It seems to me that there is a contradiction between these two sentences.  I understand that "no consent was sought", since not required by French legislation. This should  be reported uniformly.

Response 6 : We thank the reviewer for this comment. For this study, formal written consent was not required according to French legislation, but informed oral consent was obtained from all participants. This is now clearly stated in the manuscript.

  1. On line 58 one reads: "Type I interferon chemokines molecules...". Are the authors referring to IFN-induced chemokines or are they referring to the biological activity of IFN, or other. Please, clarify.

Response 7 : We thank the reviewer for this comment. We were referring to both IFN-induced chemokines and the biological activity of IFN (Skaug et al. Cytokine, 2020). We have modified the manuscript to make this clear (page 2 line 66).

  1. Importantly, in the discussion the authors should clarify what is their thinking concerning the use of the predictors they describe in the MS at the single patient level. a) On the light that the levels of the variables are most often in the normal range. b) The strengths of correlations are weak to very weak.

Response 8 : We thank the reviewer for this comment. We have modified the discussion section to address these points (page 16 line 2029).

  1. I personally think that the most useful table is the current supplementary table 6. This should go to the main text and could be improved by grading the colors according to the strength of the significance. In the case, current table 2 could go to the supplementary.

     Response 9 : We thank the reviewer for this comment. We have thus modified our tables that appears on page 12 line 1729.

  1. Line 134 "after adjustment for type of SSc". Is this adjustment or simply classification? Similarly, line 193, "p values are adjusted for type SSc." Further, line 199: "These results are shown in Table 3 and were adjusted with the type of SSc". Please, clarify.

Response 10 : We thank the reviewer for this comment. This is the adjustment according to the type of SSc and not simply a classification.

Minors

  1. The text could be improved when revised by a proficient English writer. To many are the specific terms which need intervention to allow their listing here.

Response 1 : We thank the reviewer for this comment. The manuscript has now been revised by a native English speaker.

  1. The adverb "also" is repeated 22 times in the text. Is it really necessary?

             Response 2 : We thank the reviewer for pointing out the repetition of “also”. In the revised version of the manuscript, the word is used more sparingly and only where appropriate.

Reviewer 2 Report

In this article, the authors explore the role of routine lab analysis in the clinical presentation of systemic sclerosis. The data are well presented and treated. 

In the present retrospective clinical and laboratory study, Chikhoune and colleagues explore the correlations between routine laboratory analysis and clinical presentation of systemic sclerosis. In particular, they found a significant correlation between white blood cells, diffuse cutaneous sclerosis, a correlation between CRP levels and interstitial lung disease, and a correlation between low hemoglobin and ferritin at baseline with pulmonary hypertension, among others. The introduction and discussion are adequate, and the bibliography is updated with recent articles. All the data are correctly presented. Although this article adds little to the pathophysiology of systemic sclerosis, it could represent a useful clinical tool for the rheumatologist to interpret routine blood values in clinical practice.

Author Response

In this article, the authors explore the role of routine lab analysis in the clinical presentation of systemic sclerosis. The data are well presented and treated. 

In the present retrospective clinical and laboratory study, Chikhoune and colleagues explore the correlations between routine laboratory analysis and clinical presentation of systemic sclerosis. In particular, they found a significant correlation between white blood cells, diffuse cutaneous sclerosis, a correlation between CRP levels and interstitial lung disease, and a correlation between low hemoglobin and ferritin at baseline with pulmonary hypertension, among others. The introduction and discussion are adequate, and the bibliography is updated with recent articles. All the data are correctly presented. Although this article adds little to the pathophysiology of systemic sclerosis, it could represent a useful clinical tool for the rheumatologist to interpret routine blood values in clinical practice.

  • We thank the reviewer for these comments.

Round 2

Reviewer 1 Report

None